# MicroPET Imaging Assessment of Brain Tau and Amyloid Deposition in 6 × Tg Alzheimer’s Disease Model Mice

**DOI:** 10.3390/ijms23105485

**Published:** 2022-05-14

**Authors:** ShinWoo Kang, Jinho Kim, Sang-Yoon Lee, Nobuyuki Okamura, Keun-A Chang

**Affiliations:** 1Department of Pharmacology, College of Medicine, Gachon University, Incheon 21999, Korea; swkang010@gmail.com; 2Neuroscience Research Institute, Gachon University, Incheon 21565, Korea; jinho.k.0331@gmail.com (J.K.); rchemist@gachon.ac.kr (S.-Y.L.); 3Department of Molecular Pharmacology and Experimental Therapeutics, Mayo Clinic, Rochester, VT 55905, USA; 4Gachon Advanced Institute for Health Science and Technology, Graduate School, Gachon University, Incheon 21999, Korea; 5Department of Neuroscience, College of Medicine, Gachon University, Incheon 21936, Korea; 6Division of Pharmacology, Faculty of Medicine, Tohoku Medical and Pharmaceutical University, Sendai 980-8576, Japan; nookamura@tohoku-mpu.ac.jp

**Keywords:** Alzheimer’s disease, microPET, Flutemetamol, THK5351, DPA714, MK6240

## Abstract

Alzheimer’s disease (AD) is characterized by the deposition of extracellular amyloid plaques and intracellular accumulation of neurofibrillary tangles (NFT). Amyloid beta (Aβ) and tau imaging are widely used for diagnosing and monitoring AD in clinical settings. We evaluated the pathology of a recently developed 6 × Tg − AD (6 × Tg) mouse model by crossbreeding 5 × FAD mice with mice expressing mutant (P301L) tau protein using micro-positron emission tomography (PET) image analysis. PET studies were performed in these 6 × Tg mice using [^18^F]Flutemetamol, which is an amyloid PET radiotracer; [^18^F]THK5351 and [^18^F]MK6240, which are tau PET radiotracers; moreover, [^18^F]DPA714, which is a translocator protein (TSPO) radiotracer, and comparisons were made with age-matched mice of their respective parental strains. We compared group differences in standardized uptake value ratio (SUVR), kinetic parameters, biodistribution, and histopathology. [^18^F]Flutemetamol images showed prominent cortical uptake and matched well with 6E10 staining images from 2-month-old 6 × Tg mice. [^18^F]Flutemetamol images showed a significant correlation with [^18^F]DPA714 in the cortex and hippocampus. [^18^F]THK5351 images revealed prominent hippocampal uptake and matched well with AT8 immunostaining images in 4-month-old 6 × Tg mice. Moreover, [^18^F]THK5351 images were confirmed using [^18^F]MK6240, which revealed significant correlations in the cortex and hippocampus. Uptake of [^18^F]THK5351 or [^18^F]MK6240 was highly correlated with [^18^F]Flutemetamol in 4-month-old 6 × Tg mice. In conclusion, PET imaging revealed significant age-related uptake of Aβ, tau, and TSPO in 6 × Tg mice, which was highly correlated with age-dependent pathology.

## 1. Introduction

Alzheimer’s disease (AD) is a progressive neurodegenerative disease and the most common cause of dementia [1]. AD neuropathology is characterized by the deposition of extracellular amyloid plaques. The progressive accumulation of amyloid-β (Aβ) peptide in brain regions is thought to be the causative factor but remains controversial. Abnormal aggregation of tau, which forms the intracellular aggregation of neurofibrillary tangles (NFTs) in the brain, is another major pathological hallmark of AD. Therefore, it is important to accurately and specifically target tau deposits in vivo in the brain, along with the formation of Aβ plaques.

Molecular imaging, including positron emission tomography (PET), is a promising method for the in vivo visualization of early biomarkers of AD. PET radiotracers that can selectively detect the early stages of AD are of interest in preclinical and clinical fields. [^18^F]Flutemetamol, an Aβ-specific tracer for PET, has been widely used in patients with AD [2]. It is useful for differentiating between AD patients and healthy subjects, having high specificity (96%) and sensitivity (93%) for the detection of AD, as well as high reliability in repeated testing [3,4]. 

Tau-specific PET tracers, including the [^18^F]THK family of compounds, have recently been developed [5]. [^18^F]THK5351 has a higher signal-to-background ratio than [^18^F]THK5117 because of its better binding properties and faster kinetics [5]. [^18^F]THK5351 also exhibits better pharmacokinetics, less white matter binding, and a higher target-to-reference signal than [^18^F]THK5317 [6]. These tracers are now available for clinical assessment in patients with various tauopathies, including AD, as well as in healthy individuals. However, the time course of tau aggregation and its dynamic relationship with other pathophysiological features in various tauopathies remains unclear. 

Neuroinflammation, including the activation of microglia surrounding amyloid plaques, is a characteristic pathology of AD [7]. The abundance of translocator protein (TSPO), located predominantly in glial cells, is low in normal brain tissue but markedly increased in pathological conditions associated with microglial or astrocytic activation [8]. Therefore, TSPO is considered an ideal target for neuroinflammation imaging using PET [9,10,11,12]. [^18^F]DPA714 is a promising radiotracer targeting TSPO to investigate neuroinflammation [12]. 

Transgenic mouse models of Aβ or tau deposition can provide an in vivo platform for microPET imaging to evaluate the ability of Aβ or tau tracers to track temporal and regional deposition of Aβ or tau. However, it would be ideal to compare images of [^18^F]Flutemetamol and [^18^F]THK5351 in an AD mouse model, but no previous reports have been published yet.

In our previous study [13], the 6 × Tg-AD (6 × Tg) mouse model, which expresses mutant APPswe/Ind/fl, PS1 with double FAD mutations (M146L and L286V), and tau P301L mutation, more closely represents the pathological features of AD showing age-dependent development of Aβ plaques and tau deposition in one animal model. Spatial memory deficits and microglial activation occurred in 6 × Tg mice beginning at 2 months [13]. In 6 × Tg, the neuronal loss and synaptic loss was observed in the cortex from 4 months of age and in the hippocampus from 6 months of age [13].

Here, we conducted a comparative analysis of the pathological features in the novel 6 × Tg mice and each parental line using microPET imaging analysis with a TSPO radiotracer, as well as amyloid and tau PET tracers. These results were verified using immunohistochemistry. 

## 2. Results

### 2.1. In Vivo Amyloid, Inflammation and Tau PET Imaging of 6 × Tg Model Mice

Here, we closely observed the pathological characters of mice at 2-month intervals between 2 and 8 months of age in 6 × Tg mice in a PET study using an amyloid PET radiotracer, an inflammation (TSPO) PET radiotracer, and tau PET radiotracers (Appendix A). MicroPET images were comparatively verified with postmortem histological examination of the brains of mice collected at the end of the PET scan using Aβ, phosphorylated tau, and TSPO antibodies (Appendix A).

### 2.2. In Vivo [^18^F]Flutemetamol PET Imaging Detected Aging-Related and AD-Associated Elevation in Brain Aβ

To observe the age-dependent changes in the distribution and accumulation of amyloid plaques, we investigated the brains of 2-, 4-, 6-, and 8-month-old male WT, JNPL3, 5 × FAD, and 6 × Tg mice by PET imaging study with [^18^F]Flutemetamol (Appendix A). In vivo PET images showed an excess, age-dependent elevation in whole-brain uptake of [^18^F]Flutemetamol in 5 × FAD and 6 × Tg mice from 2–8 months of age and a significant increase in whole-brain uptake, particularly in 6 × Tg mice (Appendix A). 

We analyzed the regional uptake of [^18^F]Flutemetamol by focusing on the mouse cortex and hippocampus, which are important and vulnerable brain regions in AD (Figure 1A). Standardized uptake value ratio (SUVR) values were calculated using ROI-to-cerebellum ratios from ^18^F-radioactivity data measured over 50–60 min (Figure 1B and Appendix A). When the difference in [^18^F]Flutemetamol uptake between the 2-, 4-, 6-, and 8-months of age in the cortex and the hippocampus was expressed as the abundance of WT mice, the [^18^F]Flutemetamol signal from 2- to 8-month-old 6 × Tg mice was significantly elevated compared to age-matched WT mice (Appendix A). In the cortex, there was a significant difference at 2 months between 6 × Tg (1.16 ± 0.05, * *p* < 0.05) and WT mice, but not between 5 × FAD (1.06 ± 0.03) and WT mice (Figure 1B). Two-month-old 6 × Tg mice (1.10 ± 0.03, * *p* < 0.05) also exhibited a significant increase in the hippocampus compared to age-matched WT mice (Figure 1B).

To directly assess the actual Aβ plaque burden in the brains of 6 × Tg mice, we performed 6E10 immunostaining and Thioflavin S staining in 2-month-old male WT and 6 × Tg mice (Figure 2B and Appendix A). Aβ deposition was greater in the cortex of the hippocampus of 2-month-old 6 × Tg mice than in WT mice (Figure 2B and Appendix A). These results were consistent with a previous study showing that Aβ filled most of the cortex and hippocampus of 5 × FAD mice within 2 months [14]. The [^18^F]Flutemetamol uptake was significantly associated with the number of amyloid plaque loads in the cortex (r = 0.7199, *p* = 0.0077) and hippocampus (r = 0.6645, *p* = 0.0137) of 6 × Tg mice compared with WT mice (Figure 2C). In addition, the protein levels of Aβ_1 − 42_ in the blood plasma of 2-month-old 6 × Tg mice were significantly increased compared to age-matched WT littermates (Appendix A).

### 2.3. In Vivo [^18^F]DPA714 PET Imaging Detected AD-Associated Inflammation in the Brain

TSPO is considered a promising biomarker for neurodegenerative diseases because of the correlation between TSPO overexpression and microglial activation [10]. Therefore, we performed a microPET study using [^18^F]DPA714, a PET probe that selectively and specifically examines the density of TSPO proteins, to investigate neuroinflammation in the brains of 6 × Tg mice. [^18^F]DPA714 displayed strong binding in the brains of 6 × Tg mice compared with the other groups at 2, 4, and 8 months (Figure 3A and Appendix A). In particular, 2-month-old 6 × Tg mice showed a significant increase in the cortex (1.22 ± 0.05, * *p* < 0.05) and hippocampus (1.24 ± 0.05, ** *p* < 0.01) compared with age-matched WT or JNPL3 mice (Figure 3B and Appendix A). 

To specifically examine the presence of TSPO expression in 2-month-old male WT and 6 × Tg mouse brains, a postmortem histological examination of the brains of mice collected at the end of the PET scan was conducted using a primary TSPO antibody. Upon immunohistochemical analysis, the anatomical location of TSPO protein was successfully visualized, as shown in Figure 3C. [^18^F]DPA714 uptake was significantly associated with TSPO expression in the cortex (r = 0.8751, *p* = 0.0061) and hippocampus (r = 0.7524, *p* = 0.0252) of 6 × Tg mice compared with WT mice (Figure 3D).

The pattern of [^18^F]DPA714 binding was significantly similar to the [^18^F]Flutemetamol binding pattern in the brain (cortex, r = 0.5947, *p* = 0.0414; hippocampus, r = 0.6044, *p* = 0.0018) (Figure 3E). In the visual image and SUVR analysis, the 6 × Tg group showed significant age-dependent uptake compared with the age-matched WT group in both the [^18^F]DPA714 and [18F]Flutemetamol images.

### 2.4. In Vivo [^18^F]THK5351 and [^18^F]MK6240 PET Imaging Detected AD-Associated Elevation in Brain Tau

To investigate age-dependent changes in the density of tau in the brain, 2-, 4-, 6-, and 8-month-old male WT, JNPL3, 5 × FAD, and 6 × Tg mice were injected with [^18^F]THK5351, a selective and specific PET tracer for imaging tau deposits in AD. A 1-h dynamic PET imaging was performed for each mouse brain, and the images were reconstructed. In vivo PET images revealed an excess, age-dependent elevation in whole-brain uptake of [^18^F]THK5351 in JNPL3, 5 × FAD, and 6 × Tg mice from 2–8 months of age, and a significant increase in whole-brain uptake in 6 × Tg mice (Figure 4A and Appendix A).

Again, we focused on the mouse cortex and hippocampus. The difference in [^18^F]THK5351 uptake in the cortex among the 2-, 4-, 6-, and 8-month-old mice was expressed as the abundance relative to WT mice (Appendix A). There was a significant increase in the accumulation of [^18^F]THK5351 in the cortex and hippocampus of 4-, 6-, and 8-month-old 6 × Tg mice compared with that in age-matched WT mice (Appendix A). In addition, [^18^F]THK5351 uptake in the cortex of 8-month-old 6 × Tg (1.25 ± 0.01, *** *p* < 0.001 vs. WT, #### *p* < 0.001 vs. JNPL3) mice was significantly higher than that in age-matched JNPL3 (1.20 ± 0.02) mice (Appendix A). In 4-month-old 6 × Tg mice (1.18 ± 0.04, ** *p* < 0.01 vs. WT, #*p* < 0.05 vs. JNPL3), [^18^F]THK5351 was detected at a higher abundance in the hippocampus than in age-matched WT (1.0 ± 0.02) or JNPL3 (1.2 ± 0.01) mice (Figure 4B and Appendix A).

Tau microPET imaging using [^18^F]MK6240, another tau PET probe, was performed in 4- and 8-month-old WT and 6 × Tg mice. [^18^F]MK6240 displayed strong binding in the cortical and hippocampal regions of 6 × Tg mice compared with WT mice (Figure 4A and Appendix A). ROI-to-cerebellum ratios from ^18^F-radioactivity data measured over 50–60 min were calculated for the WT and 6 × Tg mice (Figure 4B and Appendix A). The pattern of [^18^F]MK6240 binding was significantly similar to that of [^18^F]THK5351 in the brain (cortex, r = 0.6160, *p* = 0.0329; hippocampus, r = 0.3742, *p* = 0.0292) (Figure 4C).

The location of p-Tau, according to histological examination, was correlated to the region of high [^18^F]THK5351 binding on the PET images (Figure 5A). Confocal microscopy revealed p-Tau expression using an AT8 antibody in the brains of 4-month-old 6 × Tg mice compared with age-matched WT mice (Figure 5B). [^18^F]THK5351 uptake was significantly associated with the level of p-Tau expression in the hippocampus (r = 0.6614, *p* = 0.0141), but not in the cortex (r = 0.3643, *p* = 0.1131) (Figure 5C). The protein levels of p-Tau were also increased in the brain of 4-month-old 6 × Tg mice compared to age-matched WT mice (Figure 5D). 

[^18^F]THK5351 uptake was significantly correlated with [^18^F]Flutemetamol uptake in both the cortex and hippocampus of 6 × Tg mice (cortex, r = 0.9243, *p* = 0.0247; hippocampus, r = 0.9218, *p* = 0.0004) (Figure 6A). The correlation between [^18^F]MK6240 uptake and [^18^F]Flutemetamol uptake was also assessed in the cortex and hippocampus of 6 × Tg mice. [^18^F]MK6240 uptake was also significantly correlated with [^18^F]Flutemetamol uptake in the cerebral cortex and hippocampus of 6 × Tg mice (cortex, r = 0.9475, *p* = 0.0143; hippocampus, r = 0.8559, *p* = 0.0016) (Figure 6B).

In this microPET imaging study, we identified two main pathological features, Aβ plaques and tau deposition, along with inflammation in the 6 × Tg mouse brain.

## 3. Discussion

Transgenic mouse models of Aβ or tau deposition were used as in vivo platforms for microPET imaging to evaluate the ability of Aβ or tau tracers to track the temporal and regional deposition of Aβ or tau. However, certain mouse models may not be suitable for investigating the binding of at least some tau tracers based on the few in vivo microPET studies thus far. Transgenic mice with various genetic backgrounds have been associated with different pathologies, which makes it difficult to interpret overlapping study results [15].

In our previous study, Aβ aggregates were identified from 2-month-old, and PHF tau was identified from 4-month-old 6 × Tg mouse brain [13]. In addition, we confirmed that Aβ deposits are shown in the cortex and in the hippocampus of 2-month-old 6 × Tg mice brain using Thioflavin S staining (Appendix A) and immunohistochemistry using 6E10 (Figure 2B) and the protein level of Aβ_1 − 42_ was significantly increased in the blood plasma from 2-month-old 6 × Tg mice compared to their age-matched WT littermates (Appendix A). In addition, increased p-Tau is detected in the brain of 4-month-old 6 × Tg mice compared to WT using immunohistochemistry and western blot using AT8 (Figure 5B,D)

MicroPET studies have also provided data on Aβ and tau deposition in the brains of newly developed 6 × Tg mice. We used [^18^F]Flutemetamol and [^18^F]THK5351 for amyloid plaque and tau deposition, respectively, which have been recently used for AD diagnosis. Our new mouse model showed age-dependent development of Aβ plaques and tau deposition in an animal model, and each pathological symptom appeared much faster than in a single animal model [13]. As a neuroinflammatory marker, microglial activation occurs in a neuritic plaque-dependent manner in the cortex and hippocampus of 6 × Tg mice [13]. These pathological features were confirmed in this study by using microPET imaging.

Here, we monitored the pathological characteristics of mice from 2–8 months of age at 2-month intervals. We observed an overt, age-dependent elevation in the cortical and hippocampal uptake (radioactivity) of [^18^F]Flutemetamol at 2, 4, 6, and 8 months in the brains of 5 × FAD and 6 × Tg mice compared with age-matched WT mice (Appendix A). [^18^F]Flutemetamol uptake in the cortex and hippocampus was normalized to [^18^F]Flutemetamol radioactivity in the cerebellum, a reference region that lacks these PET probes in the AD mouse model. There was no difference between 5 × FAD and 6 × Tg mice in the cortex and hippocampus, but an increased tendency was observed when compared with WT or JNPL3 mice (Appendix A). These results correlate with the histochemical data in 6 × Tg mice.

TSPO is considered a promising biomarker of microglial activation concomitant to amyloid pathology [10,16]. In this study, a microPET study using [^18^F]DPA714, a TSPO PET probe, showed strong binding in the brains of 6 × Tg mice compared with the other groups at 2, 4, and 8 months, particularly in the cerebral cortex and hippocampus. This result indicates neuroinflammation in the brains of the 6 × Tg mice. In the postmortem histological examination of the brains of mice collected at the end of the PET scan, the anatomical location of the TSPO protein was successfully associated with [^18^F]DPA714 uptake in the cortex and hippocampus of 6 × Tg mice compared with that of WT mice. Moreover, the pattern of [^18^F]DPA714 binding was significantly similar to the [^18^F]Flutemetamol binding pattern in the brain of 6 × Tg mice, and both the [^18^F]Flutemetamol and [^18^F]DPA714 images showed significant age-dependent uptake in 6 × Tg mice compared with age-matched WT mice.

Interestingly, microPET imaging of [^18^F]THK5351 was significantly increased in the brains of JNPL3 and 6 × Tg mice at 4, 6, and 8 months compared to that in WT and 5 × FAD mice (Appendix A). There was no difference in [^18^F]THK5351 uptake in the cortex of JNPL3 and 6 × Tg mice, but there was a significant increase in the hippocampus of 6 × Tg mice compared with that in JNPL3 mice (Appendix A). According to histological examination, the location of p-Tau was highly correlated with the region of high [^18^F]THK5351 binding on the PET images, particularly in the hippocampus of 6 × Tg mice (Figure 5A,B). These results support the potential of [^18^F]THK5351 for in vivo tau imaging.

Recently, [^18^F]THK5351 was shown to target monoamine oxidase B (MAO-B), an off-target binding site [17,18]. Recent reports have shown that the second-generation tau PET tracer MK6240 binds selectively and specifically to one site on the NFT-rich AD brain tissue and neither binds to off-target sites nor has an affinity for amyloid plaques [19,20,21]. Therefore, the [^18^F]THK5351 signal was confirmed by using [^18^F]MK6240. As a result, the retention of [^18^F]MK6240 increased at 4 months of age in 6 × Tg mice, similar to that of [^18^F]THK5351. Based on these data, tau PET imaging of 6 × Tg mice was performed.

Taken together with these results, the pathological features of AD were identified in the novel 6 × Tg mice using microPET imaging analysis with [^18^F]DPA714, which are TSPO radiotracers, as well as amyloid and tau PET tracers. These results were highly correlated with pathological features.

## 4. Materials and Methods

### 4.1. Animals

The 6 × Tg mice were generated via crossbreeding the 5 × FAD (B6SJL) Tg mouse (The Jackson Laboratory, Bar Harbor, ME, USA) with the JNPL3 Tg mouse (Taconic Biosciences Inc., Albany, NY, USA), as previously described [13]. Briefly, hemizygous 5 × FAD transgenic mice were crossbred with JNPL3 hemizygous transgenic mice (B6SJL background), yielding animals with four different genotypes (wild-type [WT], JNPL3 +/− [JNPL3], 5 × FAD +/− [5 × FAD], and JNPL3 +/−5 × FAD +/−[6 × Tg]). Genotyping was performed on-ear biopsy DNA by polymerase chain reaction. Animals were housed in an automatically controlled environment at 22 ± 2 °C and 50 ± 10% relative humidity under a 12-h light/dark cycle with *ad libitum* access to food and water. In each experimental group, 2-, 4-, 6-, and 8-month-old, age-matched male mice were used for analysis (Figure 1A). All animal experiments were performed in compliance with the Animal Care and Use Guidelines of the Gachon University, Seoul, Korea (LCDI-2019-0114: 2 March 2020). All animal experimental protocols using microPET were approved by the Laboratory Animal Care Committee of the Neuroscience Research Institute (NRI), Gachon University (NRI-IACUC-2020-001: 24 April 2020).

### 4.2. MicroPET Studies

[^18^F]Flutemetamol and [^18^F]DPA714 were purchased from Carecamp, Inc. (Seoul, Korea). [^18^F]THK5351 was routinely prepared at the NRI (Gachon University) using a minimally modified method based on published literature [5]. Radiosynthesis was performed using a tosylate precursor provided by Tohoku University, Japan. [^18^F]MK6240 was also routinely prepared at the NRI (Gachon University) using a minimally modified method based on published literature [22].

MicroPET scanning was performed using a Siemens Focus 120 small animal PET scanner (Siemens Preclinical Solutions Inc., Erlangen, Germany) with the 60-min list mode acquisition protocol. [^18^F]-labeled PET probes (7–10 MBq) were administered via the tail vein (single bolus). Mice were anesthetized with 2% isoflurane and 98% oxygen. The dynamic list mode data were arranged into sonograms with 27 frames (6 × 3 s, 7 × 6 s, 8 × 30 s, 1 × 300 s, and 5 × 600 s) and recomposed by two iterations of two-dimensional filtered back projection, followed by 18 iterations of the maximum *a posteriori* reconstruction algorithm. Image files were calculated by region of interest (ROI) analysis using the PMOD software (PMOD Technologies LLC, Zurich, Switzerland). ROIs associated with the striatum and cerebellum were drawn on all coronal brain images guided by stereotactic coordinates. The non-displaceable binding potential, commonly used as an indication of receptor binding density, is the ratio of the peak values of the specific binding curve (SUV striatum–SUV cerebellum) to the non-specific binding curve (SUV cerebellum) at the time of the peak. The cerebellum was used as a reference region. 

### 4.3. Immunohistochemistry

After the microPET study, the mice were euthanized according to the experimental scheme (Figure 1A). The mice were anesthetized with a Zoletil and Rompun mixture (1 mg/g, i.p.) and euthanized by transcranial perfusion with saline. After immediately removing the brain from the skull, it was dissected into hemispheres. One hemisphere was fixed in 4% paraformaldehyde at 4 °C for 24 h and then dehydrated in a 30% sucrose solution for 3 d. When the tissues were dehydrated, they were frozen in molds filled with optimal cutting-temperature compounds (Sakura, Osaka, Japan). Frozen tissues were cut at a 22-μm thickness using a cryomicrotome (Cryotome, Thermo Electron Corporation, Waltham, MA, USA) and stored at 4 °C in a cryoprotectant solution (ethylene 30% and glycerol 30% in PBS). The other hemisphere was immediately frozen in liquid nitrogen and stored at −80 °C for molecular studies. 

Extracellular Aβ load, TSPO, and phosphorylated tau protein (p-Tau) expression were evaluated in the cortex and dentate gyrus of the hippocampus by immunohistochemical analyses, as described previously [13]. Briefly, brain sections were washed three times in PBS containing 0.2% Triton X-100, followed by a blocking solution (3% normal goat serum in PBS containing 0.5% bovine serum albumin and 0.4% Tween 20) for 1 h at room temperature. The sections were incubated overnight at 4 °C with primary 6E10 antibody (BioLegend, San Diego, CA, USA), TSPO antibody (Novus Biologicals, Minneapolis, MN, USA), or AT8 antibody (Thermo Fisher Scientific, Minneapolis, MN, USA). To stain Aβ plaques, the brain sections were incubated in Thioflavin S for 10 min at room temperature. The sections were washed three times and incubated for 1 h at room temperature with Alexa Fluor 555 donkey anti-mouse IgG antibody or Alexa Fluor 555 donkey anti-rabbit IgG antibody (Invitrogen, Carlsbad, CA, USA). The brain sections were then washed three times and mounted on slides using an anti-fade mounting medium (Vector Laboratories, Burlingame, CA, USA) with DAPI. Tissue specimens were obtained using a Nikon TS2-S-SM microscope (Nikon Microscopy, Tokyo, Japan) equipped with a Nikon DS-Qi2 camera. Continuous images of the cortex and hippocampus were captured in four sections, 22 μm apart (magnification ×100). TSPO or AT8 stained brain slides from each group were compared and analyzed by the ROI intensity ratio (%) using NIS-Elements software (BR 4.40.00, Nikon Microscopy, Feasterville-Trevose, PA, USA). Once the ROI was defined, fluorescence intensity was measured and red signals representing Alexa Fluor 555 within each ROI per section were converted to percentages [23].

### 4.4. ELISA

The Aβ_1–42_ protein level was measured in the plasma of the mice. The blood was isolated using heparin coated syringe and then centrifuged at 3000 rpm for 10 min at 4 °C followed by incubated for 30 min at room temperature. After add the protease inhibitor, the plasma sample was stored in deep freezer until the experiment. The ELISA was performed according to the manufacturer’s instructions (KHB3544, Invitrogen, Waltham, MA, USA) and analyzed in duplicate. The quantification was performed using a VICTOR X4 Multimode Plate Reader (PerkinElmer, Waltham, MA, USA). 

### 4.5. Western Blot

To confirm the protein level of phosphorylated tau, we performed the western blot as previously described [24]. Briefly, the frozen cortex was homogenized using radioimmunoprecipitation assay (RIPA) buffer (150 mM NaCL, 1% NP-40, 0.5% sodium deoxycholate, 0.1% SDS, 50mM Tris, pH 8.0) containing protease inhibitors (Roche Applied Science, Mannheim, Germany) and cocktail of phosphatase inhibitors (Sigma Aldrich, St. Louis, MO, USA) and centrifuged at 13,000 rpm for 10 min at 4 °C. After lysate samples were quantified using Bradford assay (Bio-Rad Laboratories, Inc., Hercules, CA, USA), loaded onto an 10% SDS-PAGE and the proteins were transferred onto a PVDF membrane. The membrane was blocked the 3% BSA in TBS-T for 1h at room temperature and then incubated with appropriate antibody for overnight at 4 °C. After three times wash in TBS-T, the membrane incubated with secondary antibody for 1 h. The protein band was detected using ECL (Millipore, Burlington, MA, USA) and BLUE X-ray film (AGFA, Mortsel, Belgium). The band quantification was performed using the Image J software v1.4.3.67. 

### 4.6. Statistical Analysis

All statistical analyses were performed using GraphPad Prism 8.4.2 (679) software (GraphPad Software Inc., San Diego, CA, USA) and outliers were removed using the outlier calculator (significance level: Alpha = 0.05) in the GraphPad Prism software. All values are expressed as mean ± standard error of the mean (SEM). Differences in the collected data between groups were analyzed using a one-way analysis of variance (ANOVA), followed by Tukey’s HSD post hoc test. Correlations were assessed using the nonparametric Spearman’s rank correlation test. Graphs show regression lines with 95% confidence intervals. Statistical significance was set at *p* < 0.05.

## Figures and Tables

**Figure 1 ijms-23-05485-f001:**
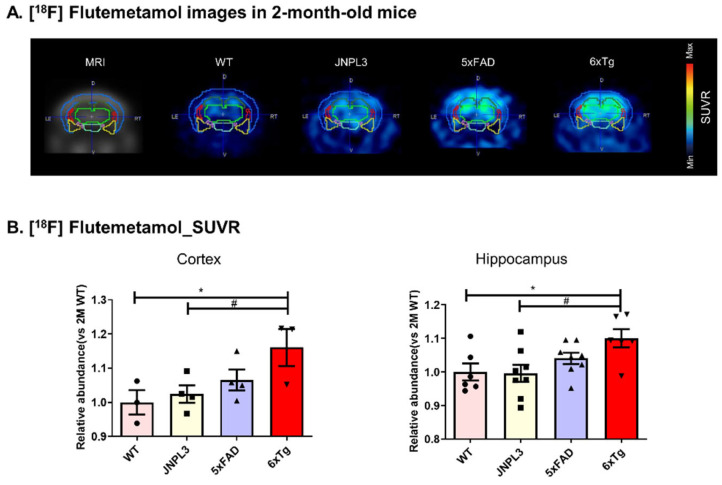
[^18^F]Flutemetamol uptake in the cortex and hippocampus of 2-month-old 6 × Tg mice brains by in vivo microPET analysis. (**A**) MicroPET images of [^18^F]Flutemetamol radiotracer in the brain. Coronal microPET images summed over 50–60 min after [^18^F]Flutemetamol injection are shown for WT, JNPL3, 5 × FAD, and 6 × Tg mice. (**B**) ROI-to-cerebellum ratios (ROI = cortex or hippocampus) from [^18^F]Flutemetamol data measured over 50–60 min were calculated for 2-month-old WT, JNPL3, 5 × FAD, and 6 × Tg mice. Increased [^18^F]Flutemetamol retention is detected in the cerebral cortex and hippocampus of 6 × Tg mice compared with that in WT or JNPL3 mice, but signals in 6 × Tg mice exhibit no significant difference compared with that in 5 × FAD mice. All data are presented as the mean ± SEM (n = 3–4 mice per group). Statistical analyses are performed using one-way ANOVA, followed by Fisher’s exact test. * *p* < 0.05 vs. WT, # *p* < 0.05 vs. JNPL3.

**Figure 2 ijms-23-05485-f002:**
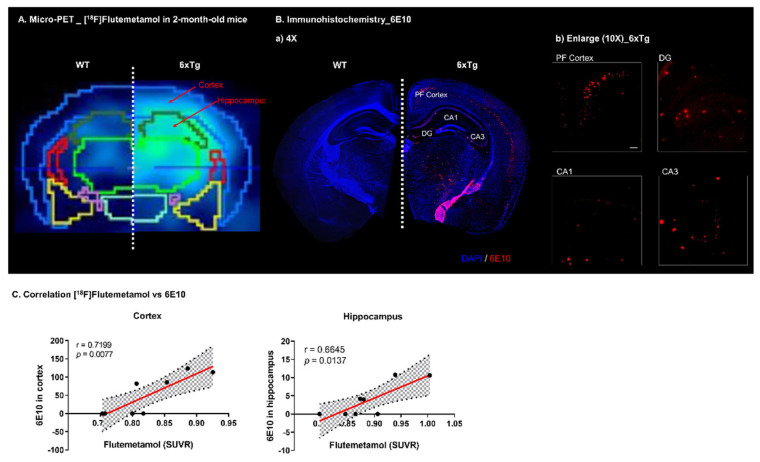
Representative images of [^18^F]Flutemetamol and 6E10 immunostained image in brain. (**A**) The [^18^F]Flutemetamol intensity of 2-month-old 6 × Tg mice increases in the cortex and hippocampus compared to that in WT mice. (**B**) Brain tissues of 2-month-old 6 × Tg and WT mice were immunostained with 6E10 antibody and counterstained with DAPI. Representative slices are shown for (a) WT and 6 × Tg mice brain (4× magnification) and (b) cortex (PF) and hippocampus (DG, CA1, and CA3) regions of 6 × Tg mice brain (10× magnification) (all scale bars = 100 μm). 6E10 stained Aβ deposits are shown in the cortex and in the hippocampus of 6 × Tg mice brain. (**C**) The correlation between [^18^F]Flutemetamol uptake and relative abundance of 6E10-stained β-amyloid deposits were assessed in the cerebral cortex and hippocampus of 6 × Tg mice by the nonparametric Spearman’s rank correlation test. Graphs show regression lines with 95% confidence intervals. [^18^F]Flutemetamol uptake significantly correlates with the relative abundance of 6E10-stained Aβ deposits both in the cerebral cortex and hippocampus of 6 × Tg mice.

**Figure 3 ijms-23-05485-f003:**
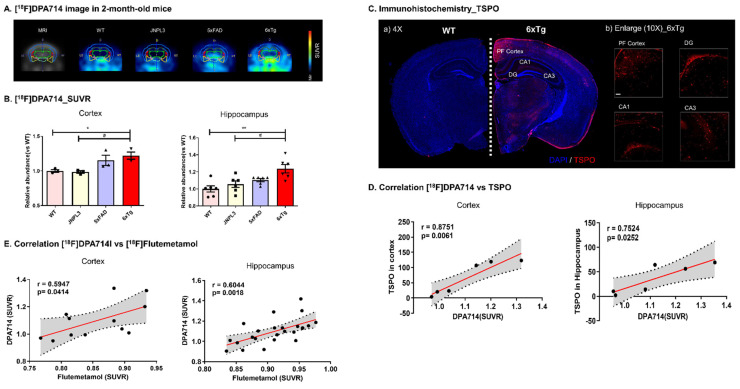
[^18^F]DPA714 microPET imaging and TSPO immunostained image in brain. Two-month-old 6 × Tg mice and their age-matched WT, JNPL3, and 5 × FAD littermates are used for the analyses (n = 3−6 per group). (**A**) Representative [^18^F]DPA714 microPET images of WT, JNPL3, 5 × FAD, and 6 × Tg mice. (**B**) ROI-to-cerebellum ratios from ^18^F-radioactivity data measured over 50−60 min are calculated for the cortical or hippocampal region in the brains of WT, JNPL3, 5 × FAD, or 6 × Tg mice. All data are presented as the mean ± SEM. Statistical analyses were performed using one-way ANOVA, followed by Fisher’s exact test. ** *p* < 0.01, * *p* < 0.05 vs. WT, # *p* < 0.05 vs. JNPL3. (**C**) Brain tissues immunostained with TSPO antibody from 2-month-old 6 × Tg and WT mouse brains. Counterstained with DAPI. Representative slices are shown for (a) WT and 6 × Tg mouse brains (4× magnification) and (b) cortex (PF) and hippocampus (DG, CA1, and CA3) regions of the 6 × Tg mouse brain (10× magnification) (all scale bars = 100 μm). TSPO stained reactive microglia are observed in the cortex and hippocampus of 6 × Tg mouse brains. (**D**) The correlation between [^18^F]DPA714 uptake and the level of TSPO protein is assessed in the cerebral cortex and hippocampus of 6 × Tg mice using nonparametric Spearman’s rank correlation test. Graphs show regression lines with 95% confidence intervals. [^18^F]DPA714 uptake significantly correlates with TSPO protein levels in both the cortex and hippocampus of 6 × Tg mice. (**E**) The correlation between [^18^F]DPA714 uptake and [^18^F]Flutemetamol uptake is assessed in the cortex and hippocampus of 6 × Tg mice using the nonparametric Spearman’s rank correlation test. Graphs show regression lines with 95% confidence intervals. [^18^F]DPA714 uptake significantly correlates with [^18^F]Flutemetamol uptake in both the cortex and hippocampus of 6 × Tg mice.

**Figure 4 ijms-23-05485-f004:**
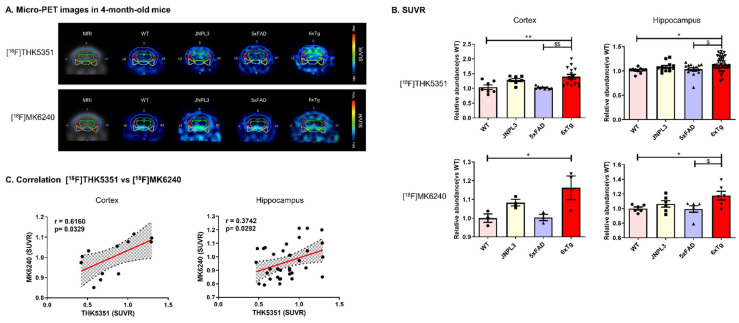
[^18^F]THK5351 and [^18^F]MK6240 uptake in the cortex and hippocampus of 4-month-old 6 × Tg mice brains by in vivo microPET analysis. (**A**) MicroPET images of [^18^F]THK5351 and [^18^F]MK6240 radiotracers in the brain. Coronal microPET images summed over 50–60 min after [^18^F]THK5351 or [^18^F]MK6240 injection are shown for the WT, JNPL3, 5 × FAD, and 6 × Tg mice. (**B**) ROI-to-cerebellum ratios (ROI = cortex or hippocampus) from [^18^F]THK5351 (a) or [^18^F]MK6240 (b) data measured over 50–60 min are calculated for 4-month-old WT, JNPL3, 5 × FAD, and 6 × Tg mice. Increased [^18^F]THK5351 or [^18^F]MK6240 retention is detected in the cerebral cortex and hippocampus of 6 × Tg mice compared with that of WT or 5 × FAD mice, but signals in 6 × Tg mice exhibit no significant difference in tau deposition compared with that in JNPL3 mice. All data are presented as the mean ± SEM (n = 3–7 mice per group). Statistical analyses are performed using one-way ANOVA followed by Fisher’s exact test. ** *p* < 0.01, * *p* < 0.05 vs. WT, ^$$^
*p* < 0.01, ^$^
*p* < 0.05 vs. 5 × FAD. (**C**) The correlation between [^18^F]THK5351 uptake and [^18^F]MK6240 uptake is assessed in the cerebral cortex and hippocampus of 6 × Tg mice using nonparametric Spearman’s rank correlation test. Graphs show regression lines with 95% confidence intervals. [^18^F]THK5351 uptake was significantly correlated with [^18^F]MK6240 uptake in both the cerebral cortex and hippocampus of 6 × Tg mice.

**Figure 5 ijms-23-05485-f005:**
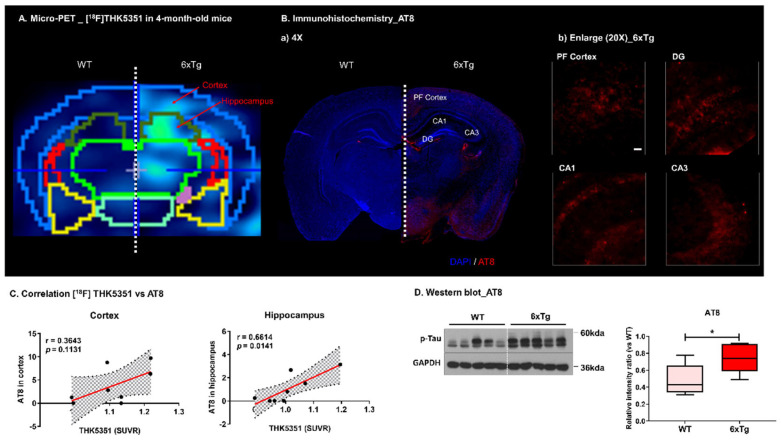
Representative images of [^18^F]THK5351 and AT8 immunostained images in brain. (**A**) The [^18^F]THK5351 intensity of 4-month-old 6 × Tg mice was increased in the cortex and hippocampus compared with WT. (**B**) Immunostained brain tissues with AT8 antibody of 4-month-old 6 × Tg and WT mice brains. Counterstained with DAPI. Representative slices are shown for (a) WT and 6 × Tg mouse brains (4× magnification) and (b) cortex (PF) and hippocampus (DG, CA1, and CA3) regions of 6 × Tg mice brain (20× magnification) (all scale bars = 100 μm). AT8-immunoreactive p-Tau were shown in the cortex and in the hippocampus of 6 × Tg mice brain. (**C**) The correlation between [^18^F]THK5351 uptake and the level of p-Tau protein are assessed in the cerebral cortex and hippocampus of 6 × Tg mice by the nonparametric Spearman’s rank correlation test. Graphs show regression lines with 95% confidence intervals. [^18^F]THK5351 uptake significantly correlates with the level of p-Tau protein in the hippocampus, but not in the cortex of 6 × Tg mice. (**D**) The protein level of phosholyated tau (p-Tau) was signficantly increased in 6 × Tg compared with WT. Statistical analysis is performed using student’s t-test. * *p* < 0.01 vs. WT.

**Figure 6 ijms-23-05485-f006:**
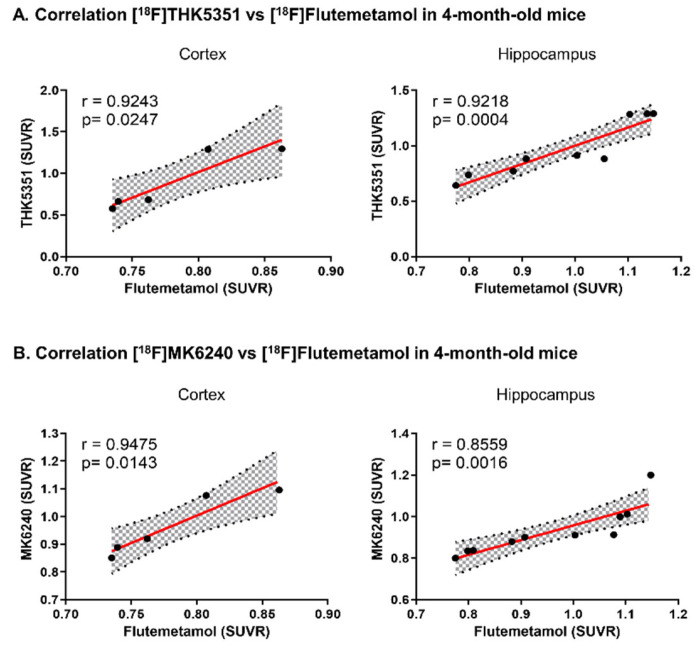
Correlation between [^18^F]THK5351 or [^18^F]MK6240 uptake and [^18^F]Flutemetamol uptake in brain. (**A**) The correlation between [^18^F]THK5351 uptake and [^18^F]Flutemetamol uptake were assessed in the cerebral cortex and hippocampus of 6 × Tg mice by the nonparametric Spearman’s rank correlation test. Graphs show regression lines with 95% confidence intervals. [^18^F]THK5351 uptake significantly correlates with [^18^F]Flutemetamol uptake both in the cortex and hippocampus of 6 × Tg mice. (**B**) The correlation between [^18^F]MK6240 uptake and [^18^F]Flutemetamol uptake are assessed in the cortex and hippocampus of 6 × Tg mice by the nonparametric Spearman’s rank correlation test. Graphs show regression lines with 95% confidence intervals. [^18^F]MK6240 uptake significantly correlates with [^18^F]Flutemetamol uptake both in the cortex and hippocampus of 6xFAD mice.

## Data Availability

The data that support the findings of this study are available from the corresponding authors upon reasonable request.

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
