# Peer review of "MicroPET Imaging Assessment of Brain Tau and Amyloid Deposition in 6 × Tg Alzheimer’s Disease Model Mice"

_ijms, 2022, doi:10.3390/ijms23105485_

Round 1

Reviewer 1 Report

This work carries out an exhaustive study about the identification of different pathological markers of AD in murine models by using PET and different radiotracers used in the diagnosis of human AD. One of the strengths of this work is the study of possible correlations between radiotracer uptake and immunoreactivity for each of the histopathological features analyzed. Unfortunately, the manuscript requires an exhaustive revision since there are numerous errors and omissions that make it very difficult to understand.

Below is a list of some of the errors.

The introduction requires mention of the characteristics of the murine models used.

Results:

  • The paragraph beginning on line 83 describes the results using a TSPO-specific radiotracer and reference is made to Figure 1. However Figure 1 describes the experimental scheme, the images of flutemetamol and the quantification of flutemetamol uptake. On the other hand, the caption of Figure 1 indicates that these are representative images of Flutemetamol and DPA714 but there is no image of DPA17.
  • On line 105 the asterisk must be before p
  • The caption of Figure 1 seems to be for Figure 2. Revise all the captions
  • On line 143, "WT mice brains. Counterstained...." spare the point
  • All the correlation graphs are too small. The labels at X and Y axes is difficult to read
  • The caption of figure 3 indicates that the figures show results for animals of 2,4 and 8 moths. Nevertheless the images are only for 2 month-old animals. The other ages are shown at the supplemental figure. This description confuses the reader since it expresses something that is not seen in the image. In addition, at line 182 it sais that brain tissues are immunostained with 6E10 Ab; this is a mistake since the panel C is stained with TSPO Ab. Also, at the line 192 appear simbols like ## and "dolar" however these symbols do not appear in the figure. In line 185 it is indicated that panel c shows the correlation between DPA714 and TSPO, but that correlation appears in panel D
  • In lines 195 and 196 the title doesn't make sense "....PET imaging detected age-related and AD-associated elevation in brain tau"

Author Response

May 4th, 2022

Dear Editor

My co-authors and I would like to thank you for allowing us to submit a revised version of our manuscript. We are grateful to the Editor and Reviewers for their positive and constructive comments and suggestions on how to improve our manuscript entitled MicroPET imaging assessment of brain tau and amyloid deposition in 6xTg Alzheimer’s disease model mice (ijms-1681728).

We have taken into consideration the critiques of the Reviewer and have revised our manuscript accordingly. Please find attached point-by-point responses to the reviewer’s comments and the revised version of our manuscript (with all changes marked in red), which we would like to submit for your consideration.

We believe that we have significantly improved the quality of our manuscript and hope it now reaches the standard of your esteemed journal, IJMS.

Once again, we would like to thank you and the reviewers for your insightful comments on our paper.

We look forward to hearing from you.

Sincerely,

Keun-A Chang, Ph.D.

Associate professor, Department of Pharmacology, College of Medicine, Gachon University

Director, Department of Basic Neuroscience, Neuroscience Research Institute, Gachon University

Reviewer 2 Report

The article presents newly developed 6xTd-AD mouse model and PET studies in these 6xTg mice using an amyloid PET radiotracer ([18F]Flutemetamol); tau PET radiotracers ([18F]THK5351 and [18F]MK6240) and translocator protein (TSPO) radiotracer ([18F]DPA714). The authors conducted a comparative analysis of the pathological features in the novel 76 6xTg mice and each parental line using microPET imaging analysis with a TSPO radio-77 tracer, amyloid and tau PET tracers. Furthermore, the microPET imaging were verified using immunohistochemistry for amyloid and tau pathology. Using a combined approach, the authors successfully showed age-dependent pathology in 6xTg mice.

Overall, the manuscript is well written, and the data is nicely presented. However, I have few concerns regarding the verification of amyloid and tau pathology. Below are my concerns/ suggestions:

1) Authors have verified the amyloid pathology using immunohistochemistry. It would be promising if authors verify Aβ(1–42) concentration in CSF and plasma using ELISA.

2) I would recommend using X34 and/or Thyoflavin S staining for Aβ plaques.

3) The western blot analysis of phosphorylated tau using pTau specific antibodies would further strengthen the data.

Author Response

May 4th, 2022

My co-authors and I would like to thank you for allowing us to submit a revised version of our manuscript. We are grateful to the Editor and Reviewers for their positive and constructive comments and suggestions on how to improve our manuscript entitled MicroPET imaging assessment of brain tau and amyloid deposition in 6xTg Alzheimer’s disease model mice (ijms-1681728).

We have taken into consideration the critiques of the Reviewer and have revised our manuscript accordingly. Please find attached point-by-point responses to the reviewer’s comments and the revised version of our manuscript (with all changes marked in red), which we would like to submit for your consideration.

We believe that we have significantly improved the quality of our manuscript and hope it now reaches the standard of your esteemed journal, IJMS.

Once again, we would like to thank you and the reviewers for your insightful comments on our paper.

We look forward to hearing from you.

Sincerely,

Keun-A Chang, Ph.D.

Associate professor, Department of Pharmacology, College of Medicine, Gachon University

Director, Department of Basic Neuroscience, Neuroscience Research Institute, Gachon University

Round 2

Reviewer 1 Report

The authors have carried out an exhaustive review and have attended to all the indications made by me. I consider that the work is of interest and can be published

Reviewer 2 Report

I am happy with the new version